# Battle of the Biomarkers of Systemic Inflammation

**DOI:** 10.3390/biology14040438

**Published:** 2025-04-18

**Authors:** Emilia Stec-Martyna, Karolina Wojtczak, Dariusz Nowak, Robert Stawski

**Affiliations:** 1Research Laboratory CoreLab, Medical University of Lodz, 6/8 Mazowiecka St., 92-215 Lodz, Poland; emilia.stec-martyna@umed.lodz.pl; 2Department of Clinical Physiology, Medical University of Lodz, 92-215 Lodz, Poland; karolina.wojtczak3@stud.umed.lodz.pl (K.W.); dariusz.nowak@umed.lodz.pl (D.N.)

**Keywords:** biomarkers, cell-free DNA (cfDNA), C-reactive protein (CRP), exercise, myocardial infarction, sepsis, systemic inflammation

## Abstract

Inflammation is a key aspect in many conditions and requires accurate biomarkers for effective assessment. While C-reactive protein (CRP) is widely used, its delayed response and low specificity limit its utility. In contrast, cell-free DNA (cfDNA) offers faster and more precise detection of cellular damage. Our study compares CRP and cfDNA as biomarkers of exercise stress, myocardial infarction, and sepsis. cfDNA rises immediately after acute injury, giving it an advantage for early diagnosis, while CRP better reflects prolonged inflammation. In sepsis, cfDNA levels strongly correlate with severity and prognosis; hence, combining both markers can improve diagnostics and treatment, supporting the role of cfDNA in modern medicine.

## 1. Introduction

Biochemical markers are popular choices for the diagnosis, monitoring, or prognosis of a condition or for assessing the response to treatment. One of the most widely used markers of inflammation is C-reactive protein (CRP). While it demonstrates consistent repeatable responses and cost effectiveness, it is commonly combined with other biochemical markers such as procalcitonin (PCT), interleukin-6 (IL-6), albumin, or fibrinogen to enhance its diagnostic accuracy and present a more comprehensive clinical picture and provide complementary information about systemic inflammation, infection, and tissue damage [1]. Moreover, CRP is often interpreted in the context of broader laboratory assessments, including complete blood count (CBC) parameters and organ function indicators such as alanine aminotransferase (ALT), aspartate aminotransferase (AST), creatinine, and urea [2]. Such combinations improve the diagnostic and prognostic value of CRP, allowing for a more accurate evaluation of inflammatory status and overall patient condition. CRP was originally identified in 1930 by Tillet and Francis as a protein whose level was elevated in patients with pneumococcal pneumonia [3]. It is currently used for diagnosing diseases, monitoring disease progression, and assessing treatment responses for inter alia infections and autoimmune or cardiovascular diseases [4].

A more recently discovered biomarker is cell-free DNA (cfDNA), which was first identified in 1948 [5]. However, cfDNA has not achieved the same widespread application as an essential marker in clinical diagnostics as CRP. Nevertheless, its popularity has steadily increased in recent years, driven by intriguing findings regarding its elevation during physical exercise and its potential applications in oncology and prenatal diagnostics [6]. The present study focuses on total circulating cfDNA, which may serve as an indicator of systemic inflammation or overall well-being, rather than on tumor-derived or fetal-specific cfDNA.

CRP and cfDNA differ significantly with regard to their origins. CRP is produced mainly in the liver in response to stimuli from the immune system, such as cytokines and other inflammatory mediators [4]. In contrast, the mechanism of cfDNA release is far more complex. It appears to originate primarily from either damaged or stimulated cells through processes such as apoptosis, necrosis, NETosis, and spontaneous release [7]. To sum up, CRP and cfDNA have differing roles and dynamics in clinical and physiological contexts.

While C-reactive protein (CRP) has long been used in the diagnosis and monitoring of inflammatory conditions, its lack of specificity limits its clinical utility. By comparison, cell-free DNA (cfDNA) has arisen as a promising biomarker, offering an understanding of cellular damage, neoplastic genetic alterations, and pathogen detection. However, very few studies have directly compared the diagnostic and prognostic value of CRP and cfDNA across various medical conditions. Such comparisons are needed to identify a biomarker or a combination of biomarkers that provide the most accurate and cost-effective approach in different clinical scenarios. By addressing this gap, our study aims to enhance the understanding of their roles, ultimately improving patient care through more precise and personalized diagnostic strategies.

Both CRP and cfDNA levels demonstrate well-established associations with systemic inflammation and tissue damage. Moreover, due to the dynamic and rapid nature of the inflammatory response observed in these diseases, this review focuses on the effect of acute conditions, such as myocardial infarction, sepsis, and physical exercise, on CRP and cfDNA levels. Indeed, acute conditions often involve a quick onset of symptoms, accompanied by significant, measurable fluctuations in biomarkers such as cfDNA and CRP; as such, they are particularly useful for real-time diagnosis, monitoring, and prognosis.

In contrast, chronic inflammation appearing in cancer, autoimmune disorders, and transplant-related complications generally has a slower, more insidious progression. The inflammatory processes in these diseases are often persistent but less dramatic in their changes over short periods, making them more difficult to study based on the rapid shifts in biomarker levels. Such complex, long-term inflammatory processes demand a more focused review dedicated to each of them to fully explore the role of cfDNA and CRP.

## 2. Comparative Analysis in Exercise

Physical exercise triggers numerous physiological adaptations across multiple organ systems. These exercise-induced changes affect cardiac output, immunological parameters, substrate metabolism, and hormonal regulation, collectively contributing to both acute responses and chronic adaptations that promote health [8].

Noteworthily, cell-free DNA (cfDNA) exhibits a rapid response to physical exercise. Its levels increase within minutes of initiating exercise, reaching a maximum immediately at the end of the activity; they then decline rapidly, probably due to DNase activity and renal clearance, with concentrations typically returning to baseline in less than 24 h. In contrast, C-reactive protein (CRP) demonstrates a markedly delayed response. The level of CRP measured immediately post-exercise increases only slightly, rising only by 20 to 30 percent. Notably, this observation can lead to “measurement bias” and raises concerns about the reliability of CRP as a marker of exercise load [9]. Its levels begin to rise after a delay of up to 24 h and may continue to increase until 48 h post-exercise, eventually reaching a noticeable peak (Figure 1) [10]. Unlike cfDNA, CRP must first be proteolytically degraded into smaller fragments—primarily in the liver—before renal elimination from circulation. Noteworthily, neither CRP nor cfDNA predominantly originates directly from the muscle tissue. Instead, cfDNA is primarily released by hematopoietic lineage cells, while CRP is synthesized in the liver [11]. These differences in kinetics emphasize their potentially different diagnostic values: while cfDNA serves as an immediate marker of cellular damage, CRP reflects the inflammatory response over a longer time.

Studies have shown that longer and more intense exercise is associated with higher levels of both cfDNA and CRP. For example, endurance activities like marathon running or prolonged cycling lead to significant increases in both biomarkers due to sustained muscle damage and systemic stress. However, shorter or moderate-intensity exercise, such as a 30 min jog, results in smaller and more transient elevations. This suggests that the duration and intensity of exercise are key determinants of the magnitude of cfDNA and CRP release [12,13,14].

Understanding the association between cfDNA and CRP in response to different types of exercise has important implications for athlete monitoring, recovery, and optimizing training protocols to enhance performance and reduce injury risks. For example, both biomarkers may be of interest to prolonged endurance athletes to assess the extent of muscle damage and inflammation, while recreational exercisers may focus more on CRP as an indicator of systemic stress. The simultaneous monitoring of cell-free DNA (cfDNA) and C-reactive protein (CRP) offers a promising approach to assessing exercise-induced stress and recovery. Our combined diagnostic strategy can be particularly valuable in preventing overtraining syndrome (OTS), a condition characterized by prolonged fatigue, decreased performance, and systemic inflammation resulting from excessive training without adequate recovery [15].

Both cell-free DNA (cfDNA) and C-reactive protein (CRP) serve as valuable biomarkers not only for assessing acute exercise-induced stress but also for highlighting the long-term positive effects of physical activity. Interestingly, their levels, when analyzed over time, can reflect the beneficial impact of regular exercise on overall health. Research suggests that individuals who engage in consistent physical activity tend to exhibit lower baseline levels of both cfDNA and CRP, which correlates with long-term reduced systemic inflammation and improved health outcomes [16]. Moreover, regular exercise has been shown to reduce chronic low-grade inflammation, a key factor in the development of many diseases, including cardiovascular disorders, diabetes, and metabolic syndrome. Lower baseline levels of CRP are commonly observed in individuals who maintain an active lifestyle. Similarly, reduced levels of cfDNA indicate less cellular damage and apoptosis, which is associated with better tissue health and repair mechanisms. Together, these biomarkers suggest that regular physical activity promotes a healthier inflammatory profile and enhanced cellular integrity. Taken together, the relationship between exercise and these biomarkers is bidirectional [11,17]. While intense or prolonged exercise can temporarily elevate cfDNA and CRP levels, regular moderate exercise leads to a long-term reduction in their baseline concentrations. This adaptation reflects the body’s improved ability to manage inflammation and repair cellular damage, which are key benefits of consistent physical activity. Beyond the relatively physiological stress of exercise, the diagnostic assessment of these markers is also essential in life-threatening conditions such as MI.

## 3. Comparative Analysis in Myocardial Infarction

Myocardial infarction (MI), commonly known as a heart attack, is a life-threatening condition characterized by the interruption of blood flow to the heart muscle, leading to myocardial cell death. Early diagnosis and accurate monitoring of cardiac infarction are critical for improving patient outcomes. Cardiac troponins remain the standard for diagnosing MI due to their high specificity and sensitivity. Other markers, such as CK-MB, myoglobin, and H-FABP, provide additional diagnostic and prognostic information, especially in the early stages [18]. In addition, secondary markers, like copeptin and natriuretic peptides, are increasingly used for risk stratification and prognosis. Although CRP is not the primary biochemical marker for myocardial infarction, it is frequently employed as an important additional indicator. Its measurement provides valuable look into the extent of the inflammatory response and can contribute to risk stratification and prognosis when used in conjunction with more specific markers of myocardial injury [19,20].

In recent years, biomarkers such as cell-free DNA and C-reactive protein have gained attention for their potential roles in assessing myocardial injury and systemic inflammation associated with cardiac infarction. Despite their distinct biological origins, the two biomarkers provide a complementary view into the pathophysiology of cardiac infarction, with cfDNA reflecting cellular damage and apoptosis and CRP being a well-established marker of systemic inflammation.

In MI, cfDNA originates from multiple sources, predominantly from neutrophils about 75%; also, about 10% derives from the left atrium, thus highlighting cardiac-specific changes. In contrast, CRP is a secondary responder produced in the liver in response to a stimulus from the immunological system, primarily as part of the acute-phase inflammatory reaction [21,22]. CRP is also produced by the liver in response to inflammatory cytokines; in myocardial infarction, it exhibits a delayed increase compared to cfDNA but persists at a higher level and for a longer duration, reflecting the ongoing inflammatory response. However, cfDNA does not increase as sharply as during physical exercise. Moreover, its decline is not as rapid as the drop observed after exercise but rather follows a slower stabilization process, seemingly reflecting the gradual recovery of the patient after the peak traumatic event (Figure 2).

Despite these differences in kinetics, both markers have been shown to correlate with clinical outcomes, including recovery and complications. Elevated levels of cfDNA and CRP are associated with larger infarct size, poorer left ventricular function, and an increased risk of adverse events such as heart failure, arrhythmias, and mortality. For example, CRP measurement can help assess the extent of the infarction and predict the risk of some complications: patients with CRP levels above 20 mg/dL have a significantly higher risk of cardiac problems and death within one year [23].

The measurement might also influence therapeutic strategies, where high CRP levels may indicate the need for more aggressive treatment, such as early revascularization or anti-inflammatory therapy to reduce the risk of complications [12]. Neither cfDNA nor CRP alone could be considered a fully specific marker of inflammation for cardiac conditions. While both are valuable inflammatory indicators, CRP alone is insufficient for definitively diagnosing heart diseases. It is essential to incorporate additional tests such as ECG; echocardiography; and, in some cases, CT scans, cardiac catheterization, or stress tests. Notably, it is important to emphasize that the lack of specificity of CRP and cfDNA alone can be advantageous, as most conditions that increase the risk of MI such as chronic inflammation, autoimmune diseases, and diabetes also tend to increase their levels [20]. While MI showcases how cfDNA and CRP complementarily reflect cardiac injury and inflammation, their roles become even more critical in sepsis, which affects the whole body.

## 4. Comparison of cfDNA and CRP in Sepsis

Sepsis, a life-threatening condition, is characterized by a dysregulated immune response to infection, leading to the spread of inflammation, tissue damage, and potential organ failure; as such, the condition requires rapid and precise diagnosis and monitoring. CRP is an acute-phase protein produced by the liver in response to inflammation. Although its levels rise in response to infections, injuries, or other inflammatory conditions, this rise is not specific to sepsis [24]. Therefore, while CRP is regarded as a primary biochemical marker for sepsis due to its rapid rise in response to inflammation, its lack of specificity limits its diagnostic value. Procalcitonin (PCT) and lactate are more specific markers: PCT indicates bacterial infection, and lactate reflects tissue hypoxia and organ dysfunction, making them critical for early diagnosis and severity assessment. Moreover, microbiological testing and imaging are improving diagnostic accuracy and guiding targeted treatment strategies for sepsis [25,26].

In recent years, cfDNA and CRP have gained attention as potential biomarkers for assessing sepsis. Although they reflect different aspects of sepsis pathophysiology, when used in combination, they can provide a deeper insight into diagnostics, prognosis, and patient management [4].

As shown in Figure 3, CRP levels rise within 6–12 h after the onset of inflammation, peaking at 24–48 h, and remain elevated as long as the inflammatory stimulus persists. However, this delayed response limits the usefulness of CRP in early sepsis diagnosis [4]. In contrast, cfDNA increases rapidly within hours of cellular damage, making it an early marker of sepsis: its levels correlate with the severity of organ dysfunction and can provide real-time information about disease progression [27,28].

In sepsis, while CRP is derived from the liver, cfDNA primarily originates from damaged host cells, such as neutrophils and macrophages; it can also amplify the inflammatory response by activating TLR receptors [28].

Although CRP is widely used as a marker of inflammation, its levels are elevated in a range of inflammatory conditions, and its low specificity limits its utility in diagnosing sepsis [4]. By contrast, cfDNA appears a more reliable diagnostic tool in this regard, insofar that it demonstrates higher sensitivity and specificity. Meta-analyses have found cfDNA levels to be significantly higher in sepsis patients than in control groups (SMD = 1.02–3.0) [29].

Elevated CRP levels have been associated with worse outcomes and higher mortality rates in sepsis, while higher cfDNA levels are strongly correlated with disease severity, organ failure, and mortality. Of the two, cfDNA seems to be a more reliable predictor of adverse outcomes than CRP [30]. Also, cfDNA seems to be efficient in early sepsis diagnosis, risk stratification, and monitoring of treatment response. Its ability to reflect cellular damage and immune activation makes it a promising tool for personalized sepsis management [2] As CRP provides information about systemic inflammation and the cfDNA level reflects cellular damage and immune activation, their combined use can enhance diagnostic and prognostic accuracy in sepsis, allowing a more comprehensive assessment of severity and progression [29,30] and ultimately leading to better patient outcomes. To sum up, both cfDNA and CRP play important roles in sepsis, but they differ in their mechanisms, diagnostic and prognostic value, and clinical applications. As such, further research is needed to establish standardized protocols for their use in clinical practice.

Besides its diagnostic function, CRP also plays a crucial role in the innate immune system and exhibits antibacterial effects. It contributes to host defense mechanisms and facilitates the clearance of apoptotic cells by binding to damaged cells and foreign pathogens. Additionally, CRP participates in the activation of the complement system via the classical pathway, enhances phagocytosis by opsonizing pathogens, and modulates the balance between pro- and anti-inflammatory responses. CRP also interacts with Fcγ receptors on immune cells, influencing cytokine production and leukocyte recruitment to sites of inflammation [31]. Additionally, CRP participates in the activation of the complement system via the classical pathway, by binding to phosphocholine residues on microbial surfaces and apoptotic cells, helping in the activation of the classical complement pathway via C1q. This enhances pathogen clearance and modulates inflammation by enhancing phagocytosis by opsonizing pathogens [32,33].

CRP, similarly, can recognize organisms and foreign molecules based on pattern recognition; noteworthily, other activators of complement such as IgG immunoglobulin cannot achieve that because these molecules only recognize distinct antigenic epitopes [33]. 

C-reactive protein binds to apoptotic cells, protects the cells from the assembly of the terminal complement components, and sustains an anti-inflammatory innate immune response mediated by TGF-α and IL10 [34]. In turn, cfDNA is a key component of NETosis, a defense mechanism in which neutrophils release extracellular traps composed of DNA and antimicrobial proteins, which cover and neutralize pathogens. Additionally, cfDNA contributes to immune activation by acting as a damage-associated molecular pattern (DAMP), triggering innate immune responses through receptors such as TLR9. This interaction promotes cytokine production, leading to the release of pro-inflammatory mediators that orchestrate the inflammatory cascade and recruit immune cells to the site of infection [35].

To summarize, both CRP and cfDNA do not just serve as biochemical markers but also actively contribute to immune defense. By taking part in pathogen clearance, immune activation, and inflammation regulation, both indirectly assist in the fight against infection.

## 5. Pathophysiology of CRP and cfDNA

While CRP is known as part of a protective immune response, its persistent elevation or excessive activation can contribute to pathophysiological processes. CRP participates in angiogenesis and atherothrombosis of cardiovascular disease. Moreover, CRP identifies and binds a variety of intrinsic ligands, which participate in the progression of cardiovascular disease. Although CRP should not be found in a healthy artery wall, it can be found in the early phases of atherogenesis, where it has been shown that this process speeds up the growth and instability of atherosclerotic plaques by encouraging endothelial dysfunction, monocyte recruitment, and pro-inflammatory signaling) [21].

CRP inhibits endothelial nitric oxide production and contributes to plaque instability by increasing endothelial cell adhesion molecule expression. Promoting monocyte recruitment into the atheromatous plaque and by enzymatically binding to modified low-density lipoprotein (LDL) [36].

CRP could induce platelet activation and thrombus growth. In addition, CRP has apparently opposing pro-angiogenic effects that determine tissue remodeling in atherosclerotic plaques and ischemic tissues. After myocardial infarction, elevated CRP levels are associated with adverse cardiac remodeling and an increased risk of heart failure. In conclusion, CRP contributes to cardiovascular disease by several mechanisms that require an in-depth analysis [37].

In septic patients, CRP can intensify tissue injury by enhancing complement and leukocyte activation. Moreover, sustained high CRP levels have been implicated in autoimmune diseases or chronic inflammation, which could drive tissue damage. Maneta et al., 2023 suggest that, while CRP serves an important immunological function, its dysregulation may contribute to disease progression and poor clinical outcomes [38].

Cell-free DNA, although initially considered just as a product of cell death, is increasingly recognized as an active player in the pathophysiology of systemic inflammation and immune dysregulation. cfDNA could be derived from neutrophil extracellular traps (NETs) but can act as a damage-associated molecular pattern (DAMP), activating pattern-recognition receptors such as TLR9 and triggering the release of pro-inflammatory cytokines. Moreover, NETs not only contribute to endothelial dysfunction but have also been found to exert procoagulant and prothrombotic activity through various mechanisms. In conditions like sepsis, high levels of circulating cfDNA contribute to endothelial damage, coagulation disturbances, and multi-organ dysfunction [39].

Furthermore, insufficient cfDNA clearance may lead to sustained immune activation. The presence of cfDNA in the circulation, especially in excessive amounts, may thus not only reflect tissue injury but actively propagate inflammatory cascades, making it both a marker and a potential mediator of pathology in acute and chronic inflammatory states [40].

Although physical exercise is usually beneficial for health, excessive physical activity can lead to pathological effects, particularly in systemic inflammation. Both cell-free DNA (cfDNA) and C-reactive protein (CRP) play significant roles in these processes, and their dysregulation can contribute to overtraining syndrome. CRP could have several pathological implications. In cases of overtraining or inadequate recovery, persistently elevated CRP levels can indicate chronic inflammation, which may contribute to long-term health risks such as the above-mentioned endothelial dysfunction, atherosclerosis, prolonged immune activation, and cardiovascular disease [41].

Also, cfDNA, by acting as a damage-associated molecular pattern (DAMP), triggers innate immune responses through receptors such as TLR9. This can lead to the release of pro-inflammatory cytokines, contributing to prolonged systemic inflammation and triggering effects similar to those caused by CRP [42].

Moreover, elevated cfDNA levels are often associated with increased oxidative stress, where released fragments of DNA can further stimulate the production of reactive oxygen species (ROS). This creates a vicious cycle of cellular damage. Surprisingly, the interaction between cfDNA and CRP during exercise is complex. Despite different origins, the pathological effects are surprisingly mutual [43].

## 6. Risk and Advantages

Standardized pre-test guidelines are needed to ensure the reliability and accuracy of biomarker measurements, especially when comparing CRP and cfDNA as markers. Pre-analytical recommendations, such as patient activity 6 h prior to sampling, are of great importance for individuals undergoing blood testing.

For instance, physical activity, such as running or exercising, shortly before blood collection, may influence cfDNA levels to a much greater extent than they influence CRP, leading to altered or unreliable results. Moreover, pre-analytical technical errors in cfDNA measurement can arise from delayed processing (at room temperature), as prolonged storage promotes leukocyte lysis and genomic DNA contamination [44,45].

Furthermore, the physiological responses to various psychological stressors can potentially alter cfDNA release patterns and clearance mechanisms, thereby introducing variability in quantitative assessments. We suspect that even minor procedural stressors like venipuncture may transiently elevate cfDNA. Such stress-induced fluctuations must be carefully considered when interpreting cfDNA data in clinical and research settings [46].

The cost of cfDNA measurement is expected to decrease significantly due to its increasing frequency of use. Additionally, cfDNA can be easily measured in unpurified serum, which simplifies its analysis and may contribute to reducing overall testing expenses.

## 7. Clinical Implications and Future Directions

In sepsis, combining CRP and cfDNA measurements could enhance early diagnosis and prognostic assessment. In myocardial infarction, monitoring cfDNA kinetics may improve the effectiveness of reperfusion therapies. During physical exertion, understanding the transient nature of these biomarkers could aid in distinguishing physiological stress from pathological conditions. Future research should focus on developing standardized protocols for measuring CRP and cfDNA kinetics across various clinical settings. Additionally, further studies should discover their potential as therapeutic targets, paving the way for more precise and personalized approaches to managing systemic inflammation (Table 1).

Our review focuses on the role of CRP and cfDNA in acute conditions, where rapid inflammatory responses require timely and reliable biomarkers for diagnosis, monitoring, and prognosis. CRP, a widely used marker of systemic inflammation, increases within 24–48 h, making it useful for assessing persistent inflammation but limiting its ability to detect early inflammatory shifts. Conversely, cfDNA, released immediately upon cellular damage, provides a dynamic and real-time reflection of tissue injury, allowing for the earlier detection of inflammatory responses. These kinetic differences highlight the complementary nature of CRP and cfDNA rather than positioning them as competing biomarkers.

The significance of CRP in acute inflammation is well established, particularly in conditions such as COVID-19 and community-acquired pneumonia (CAP), where elevated CRP levels correlate with disease severity and outcomes. In such cases, CRP concentrations are significantly reduced by corticosteroid therapy, which, in turn, is associated with decreased mortality and shorter hospital stays [47,48,49,50]. However, despite its prognostic value, CRP lacks specificity and does not provide an immediate indication of cellular damage. In contrast, cfDNA levels respond rapidly to tissue injury. In neuro-oncology, plasma cfDNA levels offer potential as a liquid biopsy marker for intracranial tumors as they have been found to vary based on tumor type and corticosteroid treatment [51].

CRP and cfDNA could also be a target of therapy. Mice lacking DNases were unable to tolerate chronic neutrophilia, quickly dying after blood vessels were occluded by NET clots. Furthermore, the damage unleashed by clots during septicemia was enhanced when these DNases were absent [52]. Meanwhile, Pepys et al., 2006 reported the efficacy of 1,6-bis(phosphocholine)-hexane as a specific small-molecule inhibitor of CRP, where injecting rats undergoing acute myocardial infarction abrogated the increase in infarct size and cardiac dysfunction produced by the injection of human CRP [44].

In clinical practice, we suggest that, in a few scenarios, combining CRP and cfDNA with other well-established biomarkers may further enhance diagnostic accuracy. For example, interleukin-6 (IL-6) is a pro-inflammatory cytokine that plays a crucial role in inducing CRP synthesis during acute inflammation. Therefore, IL-6 levels often rise earlier than CRP and could serve as an upstream indicator of immune activation. Similarly, tumor necrosis factor-alpha (TNF-α) contributes to the early inflammatory cascade and has been linked to increased cfDNA release through its effects on apoptosis and NETosis. Procalcitonin (PCT), widely used in bacterial infections and sepsis, provides additional specificity in differentiating infectious from non-infectious causes of inflammation. Those can improve the diagnostics of sepsis [53,54].

It is well known that cfDNA is a good marker of exercise load. During exercise, we have previously seen that lactate increased as a second indicator of exercise-induced stress, following cfDNA (4 times vs. 16 times). Similarly, other markers like creatine kinase or aminotransferase also could not be compared, with a very low increase. Since the percentage increments of cf n-DNA in response to exercise were many times higher than those observed for other markers, the measurement of circulating cf n-DNA could be a sensitive tool for monitoring acute exercise effects in the human body [9].

In myocardial infarction (MI), cfDNA’s early peak could complement troponin, while CRP’s delayed rise may guide post-MI anti-inflammatory therapy (e.g., colchicine in patients with high CRP).

Furthermore, cfDNA measurements have been proven to be valuable in transplant medicine, where donor-derived cfDNA levels remain stable following steroid withdrawal but increase upon steroid resumption; these levels correlate with graft rejection, immune activation, and cardiac allograft dysfunction [55].

Given these biomarkers’ distinct advantages, there is a persuasive argument for integrating CRP and cfDNA into clinical practice to enhance diagnostic accuracy and patient management in acute inflammatory conditions. CRP remains essential for tracking sustained inflammation, while cfDNA offers an early and precise indication of cellular damage, thus improving assessments of disease severity and therapeutic response. Future research should focus on standardizing cfDNA measurement techniques and discovering its role alongside CRP in various acute settings, such as sepsis, myocardial infarction, and exercise. Pre-analytical variables (such as exercise, stress, and the timing of sample collection) require careful evaluation and standardization to ensure reliable cfDNA measurements and their clinical applicability. In the future, it is essential to conduct multicenter validation studies to establish assay standardization, particularly focusing on FDA-approved procedures. These studies will play a crucial role in ensuring the reliability and reproducibility of cfDNA assays across various clinical settings. Additionally, clinical trials should be initiated to test combined biomarker algorithms to evaluate their effectiveness in improving patient outcomes. A combined approach employing both biomarkers appears to offer considerable benefits for optimizing patient care, allowing more precise risk stratification and facilitating early therapeutic interventions.

## 8. Conclusions

The dynamic fluctuations of C-reactive protein (CRP) and cell-free DNA (cfDNA) as biomarkers offer crucial insights into the pathophysiology of inter alia sepsis, myocardial infarction, and physical exertion. While both CRP and cfDNA serve as sensitive indicators of inflammation and, partly, cellular damage, their kinetics differ significantly depending on the underlying condition. Recognizing these differences is essential for improving diagnostic accuracy, prognostic assessment, and optimizing therapeutic interventions.

Standardized procedures for cfDNA measurement are desperately needed in order to realize this potential. Comparability between studies and institutions is currently limited by differences in pre-analytical factors (such as the timing of sample collection and extraction techniques) and assay platforms. Reliable implementation will depend on filling these gaps with multicenter validation studies and commonly approved kits.

## Figures and Tables

**Figure 1 biology-14-00438-f001:**
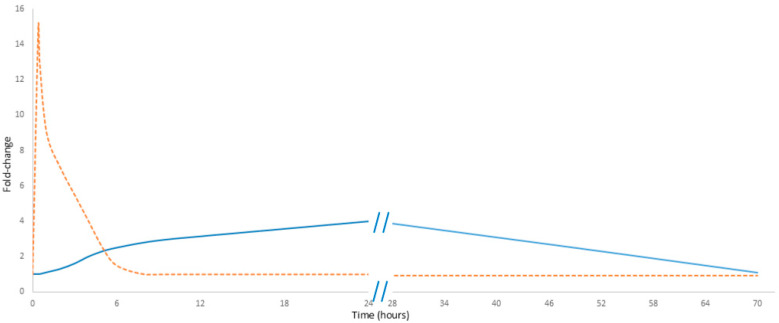
Comparative kinetics of cfDNA and CRP in exercise. The orange dashed line represents cfDNA, while the solid blue line represents CRP. Data presented in Figure 1 are detailed in Appendix A.

**Figure 2 biology-14-00438-f002:**
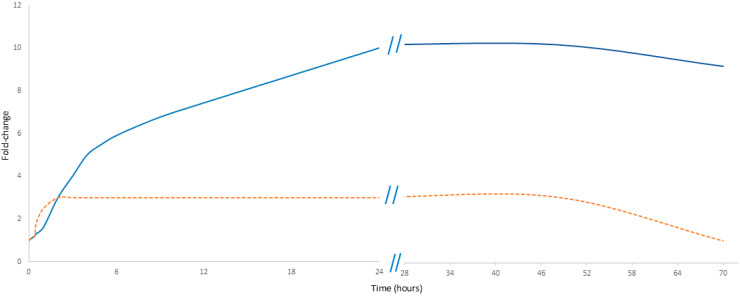
Comparative kinetics of cfDNA and CRP in myocardial infarction. The orange dashed line represents cfDNA, while the solid blue line represents CRP. Data presented in Figure 2 are detailed in Appendix A.

**Figure 3 biology-14-00438-f003:**
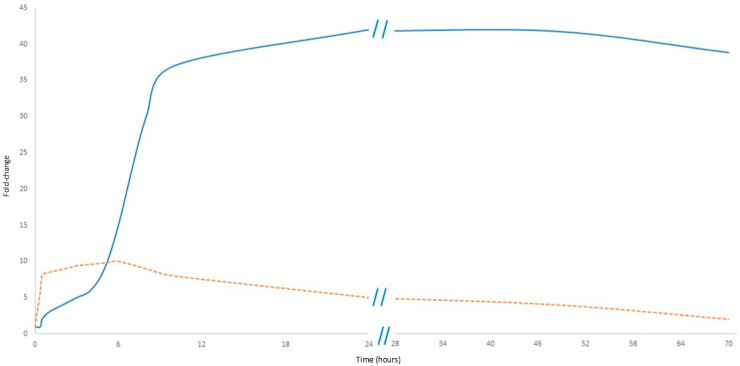
Comparative kinetics of cfDNA and CRP in sepsis. The orange dashed line represents cfDNA, while the solid blue line represents CRP. Data presented in Figure 3 are detailed in Appendix A.

**Table 1 biology-14-00438-t001:** Comparison of CRP and cfDNA as biomarkers of systemic inflammation.

Feature	CRP (C-Reactive Protein)	cfDNA (Cell-Free DNA)
Primary source	Liver (hepatocytes)	All cells; apoptotic, necrotic, activated immune cells; or NETosis
Kinetics	24–48 h after inflammation onset	Minutes to hours after tissue damage
Half-life	~19 h	~15–30 min
Specificity	Low (elevated in various inflammatory conditions)	Moderate to high (more indicative of cellular damage)
Sensitivity	High for detecting systemic inflammation	High for detecting cellular injury and immune activation
Clinical applications	Monitoring infection, sepsis, autoimmune diseases, cardiovascular risk, exercise	Early detection of sepsis, myocardial infarction, exercise
Utility in exercise	Delayed response, after 24 h average	High, well established
Utility in myocardial infarction	Delayed response, but useful marker for monitoring disease Progression	Successfully validated biomarker at the research level
Utility in sepsis	Well established	Promising early marker
Diagnostic limitations	Nonspecific; delayed kinetics limits early diagnosis	High variability, influenced by pre-analytical factors (e.g., physical activity)
Cost and accessibility	Widely available, low cost	Higher cost, but expected to decrease with broader adoption
Potential future applications	Personalized medicine (risk stratification), early detection of infections, cardiovascular risk assessment, therapeutic monitoring	Personalized inflammatory disease management, early infectious disease diagnostics (sepsis), oncology screening

## Data Availability

The data supporting the findings of this study are available upon request from the corresponding author.

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
