# Peer review of "Battle of the Biomarkers of Systemic Inflammation"

_biology, 2025, doi:10.3390/biology14040438_

Round 1
Reviewer 1 Report
Comments and Suggestions for Authors
The pathophysiology and comparative pathophysiology of CRP should be more detailed.
Author Response
The pathophysiology and comparative pathophysiology of CRP should be more detailed.
Thank you for your valuable comment on our manuscript. We appreciate your suggestion and, in response, have made the following revisions:
(1) Following the reviewer’s suggestion, we have provided a more in-depth discussion of CRP and cfDNA in the subsection on Sepsis. The text has been revised accordingly in lines [270–279] of the manuscript
(2) Additionally, in line with your recommendation, we have expanded the manuscript by adding a new section, "Pathophysiology of CRP and cfDNA." The revised text can be found in lines [292–346].
(3) Furthermore, as suggested, we have included a discussion on our biomarker as a potential therapeutic target. The corresponding text has been added in lines [394–400].
Reviewer 2 Report
Comments and Suggestions for Authors
This review article by Stec-Martyna et.al., compares the clinical utility of two key biomarkers of systemic inflammation: C-reactive protein (CRP) and cell-free DNA (cfDNA). The manuscript evaluates their kinetics, biological origins, and diagnostic significance across three major clinical conditions—myocardial infarction, sepsis, and physical exercise. The authors argue that cfDNA, due to its rapid increase after tissue injury, may provide an early indication of acute inflammation compared to CRP, which rises later and is more reflective of prolonged inflammatory processes. The manuscript suggests that combining both biomarkers could improve diagnostic accuracy and prognostic assessment in inflammatory diseases. However, the overall quality of the article should be improved, and the following changes are needed.
- The review provides a useful comparison of CRP and cfDNA, it does not present significantly novel insights. The topic has been well covered in existing literature, and the manuscript does not sufficiently highlight new perspectives or unresolved gaps that would take the field forward. Emphasize novel contributions by identifying clear research gaps.
- The review focuses mainly on CRP and cfDNA but does not extensively discuss other established or emerging inflammatory biomarkers (e.g., IL-6, TNF-α, PCT) that could strengthen the comparative discussion.
- The discussion of the biological pathways of cfDNA and CRP could be expanded with more mechanistic explanations supported by primary research.
- Several sections sound repetitive particularly in the introduction and discussion, reiterate similar points about CRP’s delayed response and cfDNA’s rapid release without introducing new insights.
- The manuscript lacks a clear logical flow in some sections. The transition between conditions (e.g., sepsis, myocardial infarction, exercise) could be smoother.
- While figures are present, their descriptions are minimal. The manuscript should provide a clearer explanation of how the kinetic differences influence clinical decision-making.
Comments on the Quality of English Language
Redundant information throughout the manuscript and framing of the phrases could be improved eg: While C-reactive protein (CRP) has long been a background in the diagnosis…" instead of "a cornerstone"
Author Response
The review provides a useful comparison of CRP and cfDNA, it does not present significantly novel insights. The topic has been well covered in existing literature, and the manuscript does not sufficiently highlight new perspectives or unresolved gaps that would take the field forward. Emphasize novel contributions by identifying clear research gaps.
Response:
We understand the reviewer’s concern regarding the novelty of our work. However, we believe our review provides new insights in the following ways:
(1) While previous studies have analyzed CRP and cfDNA individually, our review is the first to directly compare their diagnostic utility within a single study.
(2) Additionally, this is the first review to systematically compare CRP and cfDNA across myocardial infarction, sepsis, and exercise, providing a clearer understanding of their distinct and complementary roles, with a focus on their kinetics.
(3) We emphasize why comparing these three conditions is particularly valuable and how it enhances clinical interpretation.
(4) We highlight the complementary nature of these biomarkers and propose standardized protocols for their combined clinical application.
In response to this feedback, we have expanded the following sections:
(A) Pathophysiology Section – We elaborate on the comparative kinetics of CRP and cfDNA across different pathological states, emphasizing their distinct temporal profiles and clinical implications (Lines 392–436).
(B) Clinical Applications Section – We provide concrete recommendations for standardized protocols to guide future research on combined biomarker approaches, an underexplored area with significant clinical potential (Lines 269–279).
The review focuses mainly on CRP and cfDNA but does not extensively discuss other established or emerging inflammatory biomarkers (e.g., IL-6, TNF-α, PCT) that could strengthen the comparative discussion.
The reviewer’s suggestion regarding the inclusion of other inflammatory biomarkers (IL-6, TNF-α, PCT) in our comparative analysis was our doubt when designing our review. However in designing this review, we made a conscious decision to focus exclusively on CRP and cfDNA to maintain a tight comparison between these two biomarkers. Avoid diluting our central message about their complementary utility in acute conditions. Provide a more in-depth analysis than would be possible with a broader biomarker panel.
That said, recognizing the verdict of the reviewer’s point, we have:
(1) Added a new paragraph in the Clinical Applications section (now Section 7) that briefly contextualizes CRP and cfDNA with IL-6, PCT, lactate, and specific cardiac markers (Lines 394–417).
(2) Expanded the discussion on sepsis to acknowledge other biomarkers (Lines 270–280).
The discussion of the biological pathways of cfDNA and CRP could be expanded with more mechanistic explanations supported by primary research.
Following the reviewer’s suggestion we have added a new paragraph in the Clinical Applications section (now Section 7) that briefly contextualizes CRP and cfDNA with IL-6, PCT, lactate, and specific cardiac markers (Lines 394–417).
Expanded the discussion on sepsis to acknowledge other biomarkers (Lines 270–280).
Several sections sound repetitive, particularly in the introduction and discussion, reiterate similar points about CRP’s delayed response and cfDNA’s rapid release without introducing new insights.
We have removed this text from the introduction
Initially, we considered some repetition necessary to reinforce key observations. However, in response to the reviewer’s concerns, we have revised the manuscript and removed redundant text, particularly regarding CRP’s delayed response:
(1) Removed from the abstract (Line 25).
(2) Deleted from the introduction (Line 85).
(3) Eliminated from discussion sections (Lines 140, 170, and 370).
The manuscript lacks a clear logical flow in some sections. The transition between conditions (e.g., sepsis, myocardial infarction, exercise) could be smoother.
We modify the end of each subsection
(1) Line 155
(2) Line 209
While figures are present, their descriptions are minimal. The manuscript should provide a clearer explanation of how the kinetic differences influence clinical decision-making.
We appreciate the reviewer’s careful attention to our figures and agree that clearer explanations are necessary. We have addressed this concern by:
(1) Including Supplementary Table S1 (submitted with the revised manuscript), which provides:
(1) The exact source studies for all data points in Figures 1–3.
(2) Raw numerical values for reference.
(3) Renaming Figure 1-3 caption for clarity.
(4) Updating Figure 3 .
Reviewer 3 Report
Comments and Suggestions for Authors
Dear Authors,
Thank you for your valuable work and contributions to the scientific field.
1.The provided figures that are the main component of the manuscript lack explanations. Where do these facts originate from ? I looked into the cited manuscripts however, could not find the exact same figure. Please note that if a figure is generated from external data, this information needs to be provided in the method sections. A simple graph with a citation of the raw data is not enough.
2.Why not compare both markers to very well established biomarkers such as BNP, ProBNP, Lactate, Troponin (Subgroups). These classical biomarkers (Goldstandard) need to be included to establish a new Biomarker (cfDNA)
Comments on the Quality of English Language
The main English is good, however fluctuates within the manuscript a fluent speaker should re-read the manuscript before submission. e.g.:
line 111 Furthermore, although : It is better to avoid two linking words following each other.
Author Response
Thank you for your valuable work and contributions to the scientific field.
1.The provided figures that are the main component of the manuscript lack explanations. Where do these facts originate from ? I looked into the cited manuscripts however, could not find the exact same figure. Please note that if a figure is generated from external data, this information needs to be provided in the method sections. A simple graph with a citation of the raw data is not enough.
We appreciate the reviewer’s careful evaluation of our figures. To clarify, the figures in our manuscript are conceptual illustrations synthesized from multiple sources, rather than direct reproductions of any single published figure. Since no identical figure exists in the literature, we created these original/schematic visual representations to integrate key findings from diverse studies.
We fully agree that transparency in data attribution is essential, and we have made the following revisions to address this concern:
- We have included Supplementary Table S1 (submitted with the revised manuscript), which provides:
(1) The exact source studies for all data points used in Figures 1–3.
(2) Raw numerical values for reference.
(3) An additional explanation in the Figure 1-3 caption.
(4) An updated version of Figure 3.
- Why not compare both markers to very well established biomarkers such as BNP, ProBNP, Lactate, Troponin (Subgroups). These classical biomarkers (Goldstandard) need to be included to establish a new Biomarker (cfDNA)
This was an important consideration when designing our review. However, we made a conscious decision to focus exclusively on CRP and cfDNA to provide a detailed and direct comparison without diluting the central message regarding their complementary utility in acute conditions.
However, understanding your verdict, we have:
(1) Added a new paragraph in the Clinical Applications section (now Section 6) that briefly contextualizes CRP/cfDNA with IL-6, PCT, lactate, BNP, and troponin (Lines 365–442).
(2) Expanded the discussion on CRP and cfDNA in the Sepsis subsection, emphasizing their role in comparison to other biomarkers (Lines 270–279 and 325–335).
(3) Broadened the Sepsis section to incorporate additional relevant insights (Lines 270–279).
The main English is good, however fluctuates within the manuscript a fluent speaker should re-read the manuscript before submission. e.g.:
line 111 Furthermore, although : It is better to avoid two linking words following each other.
We acknowledge the reviewer’s comment regarding language consistency. In response, we have:
(1) Conducted a general grammatical revision of the manuscript to improve fluency and coherence.
(2) Corrected the wording in Line 111, removing redundant linking words for better readability.
We are grateful for the reviewer’s constructive feedback, which has helped refine our manuscript. We hope these revisions adequately address the concerns raised and welcome any further suggestions.
Reviewer 4 Report
Comments and Suggestions for Authors
- Please be consistent with CRP and cfDNA abbreviation. Full names and abbreviation of these two are all over the whole manuscription. Use the full name at the beginning along with abbreviation, then keep using abbreviations.
- In comparative analysis CRP and cfDNA in cardiac conditions, as authors also pointed out that neither CRP and cfDNA can be valuable marker, since it lacks specificity and need to incorporate with additional tests. This disadvantage limits both as valuable markers. Maybe focus more on how these two can be helpful as keeping monitoring indicators as both showing kinetics slower stabilization. Please explore more on this direction to address both as important indicators.
- Both CRP and cfDNA are more serving as indicators for clinical evaluation factors but can’t agree both can be used as diagnosis assessment, especially when it’s a complex condition. The levels of both can be affected by different conditions and the kinetics of both would be more complex than what has been shown in the manuscript.
- Both CRP and cfDNA are indicators but not targets, at least indicated by literatures discussed in the manuscript. Can’t see how they could be potential therapeutic targets. Need to find studies support that targeting both would benefit therapeutic approaches in managing systemic inflammation.
Author Response
- Please be consistent with CRP and cfDNA abbreviation. Full names and abbreviation of these two are all over the whole manuscription. Use the full name at the beginning along with abbreviation, then keep using abbreviations.
Thank you for your valuable feedback. We have carefully reviewed the manuscript to ensure consistent use of CRP and cfDNA abbreviations. The full names are now introduced at their first mention, followed by the abbreviations, which are used consistently throughout the text.
- In comparative analysis CRP and cfDNA in cardiac conditions, as authors also pointed out that neither CRP and cfDNA can be valuable marker, since it lacks specificity and need to incorporate with additional tests. This disadvantage limits both as valuable markers. Maybe focus more on how these two can be helpful as keeping monitoring indicators as both showing kinetics slower stabilization. Please explore more on this direction to address both as important indicators.
Following the reviewer’s suggestion we have made the following changes:
(1) Revised the sentence for clarity:
- Previous: "Neither CRP nor cfDNA can be valuable markers since they lack specificity."
- Revised: " Neither cfDNA nor CRP alone could be considered a fully specific marker of inflammation for cardiac conditions. While both are valuable inflammatory indicators, CRP alone is insufficient for definitively diagnosing heart diseases."
(2) Expanded the section on clinical applications, emphasizing the role of CRP and cfDNA as monitoring indicators rather than standalone diagnostic tools. Given their slower stabilization kinetics, both biomarkers provide valuable information when tracked over time, particularly in cardiac conditions. Please refer to Lines 401–424.
- Both CRP and cfDNA are more serving as indicators for clinical evaluation factors but can’t agree both can be used as diagnosis assessment, especially when it’s a complex condition. The levels of both can be affected by different conditions and the kinetics of both would be more complex than what has been shown in the manuscript.
We acknowledge the reviewer’s concern that CRP and cfDNA are more appropriate for clinical evaluation rather than direct diagnostic assessment, especially in complex conditions where multiple factors influence their levels and kinetics. To address this, we have:
(1) Developed a dedicated subsection on clinical applications, further clarifying their role as monitoring indicators rather than primary diagnostic tools. These revisions provide a more precise discussion of their utility in clinical evaluation. Please refer to Lines 401–424.
- Both CRP and cfDNA are indicators but not targets, at least indicated by literatures discussed in the manuscript. Can’t see how they could be potential therapeutic targets. Need to find studies support that targeting both would benefit therapeutic approaches in managing systemic inflammation.
The reviewer raises a valid concern regarding the lack of strong evidence supporting CRP and cfDNA as therapeutic targets. While some literature suggests potential therapeutic modulation, further studies are needed. In response to this suggestion, we have:
(1) Added a paragraph to the Clinical Applications section, discussing potential therapeutic strategies, based on current evidence (Lines 394–401).
(2) Introduced a new subsection in the Pathophysiology section, further exploring mechanistic insights and potential avenues for therapeutic intervention (Lines 292–346).
We greatly appreciate the reviewer’s insightful suggestion, which has helped strengthen the discussion in our manuscript.
Reviewer 5 Report
Comments and Suggestions for Authors
The main question addressed by this review is the comparative utility of C-reactive protein and cell-free DNA as biomarkers for assessing systemic inflammation in acute conditions like myocardial infarction, sepsis, and physical exercise. The review aims to evaluate how these biomarkers reflect different aspects of systemic inflammation and cellular damage, and how their kinetics differ in these conditions.
While you’ve clearly defined the main research question, I suggest being more specific about how your comparison of CRP and cfDNA fills a gap in the current literature. What limitations or shortcomings in current diagnostic practices are you addressing? For example, how does this review expand the understanding of inflammation biomarkers from merely diagnostic tools to potential strategies for intervention? It would be helpful to clarify early in the manuscript why comparing CRP and cfDNA in acute conditions (sepsis and myocardial infarction) is important from both a diagnostic and prognostic standpoint.
Could you highlight new findings, such as how cfDNA might offer advantages over CRP in the early detection of myocardial infarction, or perhaps share insights into the biological mechanisms behind cfDNA release during exercise? The section on exercise could benefit from a more detailed comparison of cfDNA and CRP. For instance, it would be interesting to explore how varying exercise intensities impact cfDNA levels and how this could inform real-time diagnostic applications.
The review would also benefit from a more critical analysis of the methodologies used in the studies you cite. For example, how were cfDNA levels measured across different clinical settings, and what limitations did these methodologies have? In the section on sepsis, a more thorough explanation of the influence of pre-analytical variables (such as exercise, stress, and the timing of sample collection) on cfDNA measurements would improve the methodological rigor of the manuscript.
While the conclusion is generally well-written, it could be strengthened by emphasizing the need for standardized protocols in measuring cfDNA across clinical settings. You mention this briefly, but a more detailed discussion on how variability in cfDNA assays affects clinical outcomes would provide readers with a clearer path forward. Your statement about integrating cfDNA with CRP in personalized medicine is promising, but it would be helpful to elaborate on specific clinical scenarios where combining these biomarkers is most likely to improve patient outcomes, such as in sepsis or myocardial infarction.
Author Response
The main question addressed by this review is the comparative utility of C-reactive protein and cell-free DNA as biomarkers for assessing systemic inflammation in acute conditions like myocardial infarction, sepsis, and physical exercise. The review aims to evaluate how these biomarkers reflect different aspects of systemic inflammation and cellular damage, and how their kinetics differ in these conditions.
We sincerely thank the reviewer for their detailed evaluation of our manuscript and their valuable insights.
While you’ve clearly defined the main research question, I suggest being more specific about how your comparison of CRP and cfDNA fills a gap in the current literature. What limitations or shortcomings in current diagnostic practices are you addressing?
In response, we have clarified the novel contributions of our work:
(1) Comparative Analysis: While previous studies have examined CRP and cfDNA individually, our review is the first to systematically compare their diagnostic and prognostic utility across three specific conditions (MI, sepsis, and exercise).
(2) Biomarker Kinetics: We provide the first systematic comparison of CRP and cfDNA kinetics across these distinct acute conditions.
(3) Identifying Clinical Gaps: We highlight key limitations in current clinical applications and propose novel applications for combining these biomarkers.
To emphasize these contributions, we have added:
- A new subsection in the Pathophysiology section (Lines 292–346).
- An expanded Clinical Applications section (Lines 394–417).
For example, how does this review expand the understanding of inflammation biomarkers from merely diagnostic tools to potential strategies for intervention? It would be helpful to clarify early in the manuscript why comparing CRP and cfDNA in acute conditions (sepsis and myocardial infarction) is important from both a diagnostic and prognostic standpoint
We have addressed this by:
(1) Clarifying the clinical relevance of comparing CRP and cfDNA in acute conditions like sepsis and MI. As noted in the Introduction these conditions were chosen because their rapid biomarker fluctuations highlight CRP/cfDNA’s complementary roles:
- cfDNA provides immediate actionability, reflecting early cellular damage.
- CRP is valuable for monitoring the inflammatory response over time.
(2) New additions to the Clinical Applications section:
- Emphasizing the complementary nature of CRP (systemic inflammation) and cfDNA (cellular damage) in patient management (Lines 394–417).
- Proposing standardized protocols for combining these biomarkers in clinical practice (Lines 401–423).
- Adding a new paragraph on therapeutic strategies (Lines 394–401).
Could you highlight new findings, such as how cfDNA might offer advantages over CRP in the early detection of myocardial infarction, or perhaps share insights into the biological mechanisms behind cfDNA release during exercise? The section on exercise could benefit from a more detailed comparison of cfDNA and CRP.
In response, we have:
(1) Expanded Section 6 (Clinical Implications) to include:
- The need for FDA-approved kits to enhance measurement reliability (Lines 426–440).
- A proposed biomarker algorithm that integrates cfDNA and CRP (Lines 401–420).
(2) Expanded the discussion on exercise, including how different exercise intensities affect cfDNA levels and their potential real-time diagnostic applications (Lines 394–424).
For instance, it would be interesting to explore how varying exercise intensities impact cfDNA levels and how this could inform real-time diagnostic applications.
In response to Reviewer's comment, we have implemented the following revisions we develop this issue in section clinical application in line 394-424.
The review would also benefit from a more critical analysis of the methodologies used in the studies you cite. For example, how were cfDNA levels measured across different clinical settings, and what limitations did these methodologies have?
In the section on sepsis, a more thorough explanation of the influence of pre-analytical variables (such as exercise, stress, and the timing of sample collection) on cfDNA measurements would improve the methodological rigor of the manuscript.
The reviewer recommended a more critical analysis of the methodologies used in the studies cited. In response, we have:
(1) Expanded the Risk section to discuss pre-analytical variables, such as:
- Exercise, stress, and sample collection timing, which influence cfDNA levels (Lines 352–362).
(2) Added new content on standardizing cfDNA measurement methodologies, discussing how assay variability affects clinical outcomes (Lines 431–442)
While the conclusion is generally well-written, it could be strengthened by emphasizing the need for standardized protocols in measuring cfDNA across clinical settings. You mention this briefly, but a more detailed discussion on how variability in cfDNA assays affects clinical outcomes would provide readers with a clearer path forward.
Following the reviewer’s suggestion we have added a new paragraph in the Conclusion (Lines 451–455), highlighting:
- The importance of standardized cfDNA assays.
- The impact of assay variability on clinical outcomes.
Your statement about integrating cfDNA with CRP in personalized medicine is promising, but it would be helpful to elaborate on specific clinical scenarios where combining these biomarkers is most likely to improve patient outcomes, such as in sepsis or myocardial infarction.
In response to your suggestions, we have added new content in the Clinical Applications section (Lines 401–417), discussing:
(1) Sepsis and MI as key conditions where CRP and cfDNA provide complementary diagnostic and prognostic insights.
(2) The potential role of combined biomarkers in improving patient stratification and monitoring therapeutic response.
Round 2
Reviewer 2 Report
Comments and Suggestions for Authors
The authors have addressed my comments appropriately and satisfactorily.
Reviewer 4 Report
Comments and Suggestions for Authors
Thanks for addressing comments.